# Influence of the Backward Fall Technique on the Transverse Linear Acceleration of the Head during the Fall

**DOI:** 10.3390/s23063276

**Published:** 2023-03-20

**Authors:** Andrzej Mroczkowski, Redha Taiar

**Affiliations:** 1Department of Sports and Health Promotion, University of Zielona Góra, 65-417 Zielona Góra, Poland; 2MATériaux et IngénierieMécanique (MATIM), Université de Reims Champagne Ardenne, 51100 Reims, France

**Keywords:** falls, injury prevention, biomechanics of a fall, kinesiology, martial arts, sport, health education, public health, ergonomics

## Abstract

Background: The formation of large accelerations on the head and cervical spine during a backward fall is particularly dangerous due to the possibility of affecting the central nervous system (CNS). It may eventually lead to serious injuries and even death. This research aimed to determine the effect of the backward fall technique on the linear acceleration of the head in the transverse plane in students practicing various sports disciplines. Methods: The study involved 41 students divided into two study groups. Group A consisted of 19 martial arts practitioners who, during the study, performed falls using the side aligning of the body technique. Group B consisted of 22 handball players who, during the study, performed falls using the technique performed in a way similar to a gymnastic backward roll. A rotating training simulator (RTS) was used to force falls, and a Wiva^®^ Science apparatus was used to assess acceleration. Results: The greatest differences in backward fall acceleration were found between the groups during the buttocks’ contact with the ground. Larger changes in head acceleration were noted in group B. Conclusions: The lower changes in head acceleration obtained in physical education students falling with a lateral body position compared to students training handball indicate their lower susceptibility to head, cervical spine, and pelvis injuries when falling backwards as caused by horizontal force.

## 1. Introduction

The World Health Organization defines a ‘fall’ as an event that results in a person coming to rest inadvertently on the ground, floor, or other lower level [1,2]. A fall may lead to serious injuries and may even be fatal. The best solution is thus to avoid it, and there are various types of measures that can be implemented to prevent it. Numerous researchers are preoccupied with the issue of preventing falls in terms of improving mobility as well as eliminating external factors that increase its risk [3,4]. In order to analyse reactions to fall-generating forces, accelerating treadmills are usually employed to cause experiment participants to lose their balance [5]. Platforms [6] and foot-clamps [7,8] may also be used to that purpose.

Obviously, due to external factors, a fall may be unavoidable despite adequate mobility. Practicing some professions increases the risk of a fall, e.g., in the case of employees of uniformed services, such as the police, military, and fire brigade. At the same time, the risk of a fall is increased when practicing certain sports, especially those involving frequent and direct contact between players, as in rugby, handball, football, or in martial arts. Some researchers report that perfecting proper motor habits during a fall may reduce its unwanted effects [9,10,11]. This is possible, naturally, if the forces causing a fall allow for the performance of certain motor activities. Researchers emphasize that school education should be involved in shaping appropriate motor habits that are useful during falls. However, as is evident from scientific reports, most school education does not adequately address this issue [12,13,14]. Such skills, however, are shaped by some extracurricular activities in sports clubs. Classes in martial arts such as aikido, jujitsu, aikijitsu, and judo deserve a special mention here. The condition for being admitted to sports competitions or performing self-defence techniques in these disciplines is mastering appropriate fall techniques by the participants [10,15]. The most commonly employed taxonomy involves dividing falls in terms of their direction, i.e., into forward and backward falls [16]. In terms of the direction of a fall, falling backwards is generally classified as more dangerous [17]. Oftentimes, hitting one’s head against the ground as a result of such a fall may lead to serious damage or even death [4]. Falls may also lead to severe injuries of the cervical spine [18,19], which is closely linked to the head in the biokinematic chain. According to some scientific publications, practicing sports such as handball or martial arts, such as aikido or jujitsu, can reduce the risk of hitting the head against the ground when falling backwards [10,11,14].

This article’s research was only concerned with falling backwards. In the description section, the distinction between the technique of a fall performed in a way similar to a gymnastic backward roll and a fall performed backward with the side aligning of the body was made [9,20,21]. The biomechanical correctness of these techniques is confirmed by some of the scientific studies [10,14]. Scientists attempt to create the conditions suitable for diagnosing movement habits during a fall. As a fall in real conditions may be detrimental to health, it is necessary to create conditions under which motor habits could be tested without exposing subjects to injuries. This is why non-apparatus tests were developed to study movement habits when falling backwards. Previously designed tests were developed for a fall technique similar to a gymnastic backward roll [9,22,23,24]. They are relatively uncomplicated to carry out, but the disadvantage is that the tested falls are not induced by an external force. The use of a rotating training simulator (RTS), forcing a fall using inertia, seems to be a better solution. The device is capable of inducing various types of falls [21,25,26]. Its design limits its uses to examining adults with high levels of physical fitness: it is intended mainly for people for whom falls are a frequent event as a result of a sports discipline or work performed, e.g., in uniformed services.

As shown by scientific reports [27] in which there is no significant division of the type of fall, trauma to the cervical spine is much more likely to occur with a pelvic injury than with a head injury. These somewhat surprising reports are explained, for example, by the fact that the head may act as a cushion and buffer, effectively dissipating the energy that would otherwise have been transferred to the cervical spine, thus reducing the risk of injury. However, a more plausible explanation is that the cause of the injury is due to the inertial forces generated during pelvic injury due to the head–torso connection. Such an explanation would be biomechanically justified when analysing the impact of the buttocks against the ground in the case of a backward fall [14]. Generating large inertial forces on the cervical spine is particularly dangerous due to the possibility of affecting the central nervous system (CNS). This risk is particularly high in people with Arnold-Chiari disease [18,19] or the elderly. Falls are the most common cause of cervical spine fractures in the elderly. This risk is especially increased in patients with hyperostotic conditions of the spine [27,28].

The biomechanical analysis of the backward fall shows that the moment of contact with the ground during a backward fall with an inappropriate fall technique may cause the head to hit the ground as well as generate large forces acting on the head [14]. The acceleration achieved by the head in the sagittal plane is mainly responsible for the backward or forward motion of the head when falling backward, revealing more about the forces acting horizontally towards the head [11]. The acceleration obtained in the transverse plane provides more information about the forces generated in the vertical direction relative to the head.

In order to understand it better, it is necessary to extend the considerations of Mroczkowski, who presented the biomechanical analysis of the moment of force acting on the head during a backward fall [14]. Based on his assumptions, this article presents an analysis of the relationship between the indications of the sensor mounted on the head of the falling person (Figure 1b), in terms of the direction and sense of the linear acceleration vectors in the transverse and sagittal planes, and these features of the force vectors acting on the head. Using the F = ma relation resulting from Newton’s second law of motion, it is obvious that the direction and sense of the force and acceleration vectors are the same, with the values being directly proportional. At the same time, the sensor, due to its stable attachment and negligible mass, can be considered as one solid body with the head.

Mroczkowski concluded that the torso and head may be considered two links of the kinematic chain, recognizing the connection between them as articulated (Figure 1), which, in the event of a moment of force, causes the head to move [14]. For this reason, it is assumed that there is a vector **P** corresponding to the weight of the head. If all the bodies were treated as solid during the collision, the considerations would assume that the **W** and **R** vectors should differ only in their directions. At the same time, by projecting the vector W onto the line connecting the designated point of contact of the buttocks with the ground and the centre of gravity of the head, the **N** vector (Figure 1b) is obtained (marked with a dashed line, as an exception). The N vector should also be equal to the resultant vector that would be created from the addition of vectors **P** and **Y’**. The **N** vector should only differ from the **Y** vector in sense. The same relationships will occur if the sensor indications are analysed in the other settings during the fall (Figure 1a,c,d). Some discrepancies from the relationships described above may result from the fact that the value of the **P** vector was adopted without the precise determination of the head mass. Its length may therefore be inconsistent with reality. In reality, the change of the other vectors marked in the centre of gravity of the head may be affected by the fact that not all the energy during the impact with the ground will be transferred to the head [14]. The above analysis suggests that vector **X’** will be closest to the direction of linear acceleration in the sagittal plane **a_S_**, and vector N will be closest for the transverse plane **a_T_**. This analysis shows that the indications of **a_T_** acceleration are the most informative in terms of the risk of the head, neck, spine, and pelvic injury as a result of the connection with the N force. On the other hand, the **a_s_** indicates the risk of hitting the head during a fall due to the connection with the **X’** force. Simultaneously, it should be noted that the discussed accelerations indicated by the sensor may be affected by the participation of the head in the curvilinear movement of gravitational fields. It is difficult to find scientific studies on the subject of falling backwards. Moreover, the accelerations achieved may be influenced by the muscle force generated by the players and acting on the head during the fall.

The biomechanical analysis presented in Figure 1 shows that, when the buttocks hit the ground, the main component of the force acting on the pelvis during the injury is pointed upwards (Y), with the smaller component (X) directed forwards or backwards. In the Young–Burgess classification of pelvic fractures [28,29], it is the vs. (Vertical Shear) type. With this type of fracture, the pubic and ischial bones limiting the obturator foramen may be fractured, which carries the risk of damage to the vascular and nervous structures passing through this hole and damage to the lesser pelvic organs.

The age group particularly at risk of fractures, including pelvic fractures, are the elderly, who suffer from a decrease in bone mineral density and weakening of the mechanical strength of the bones associated with osteoporosis. Pelvic fractures due to a backward fall can be classified as low-energy fractures, particularly characteristic of the elderly population. According to scientific reports, regardless of the classification of fall types, falls among people over 65 years of age are the most common cause of such fractures [30,31]. There is an increase in the incidence of low-energy pelvic fractures in people with osteoporosis [30,32]. Osteoporotic pelvic fractures in the elderly result in an annual mortality rate of 9.5% to 27% among the elderly [33].

In fact, the resultant linear acceleration vector obtained from acceleration vectors in the sagittal and transverse planes are responsible for the susceptibility of the head to damage during a fall. In order to fully analyse this susceptibility, it is necessary to examine the acceleration in both the transverse and sagittal planes, which has already been studied [11]. The results obtained in this publication concern the analysis of acceleration in the transverse plane; however, for a complete analysis, these results should be compared with the previous results concerning acceleration in the sagittal plane.

The aim of the research undertaken in this article was to investigate changes in the linear acceleration of the head in the transverse plane during a backward fall forced on the rotating trainer among representatives of various sports disciplines. These included students of physical education practicing martial arts performing falls with the side aligning of the body technique and handball players performing falls applying the technique similar to a gymnastic backward roll. At the same time, the comparison of these results with those obtained previously in the sagittal plane was attempted. The aim of the research was furthermore to check whether the values of the obtained accelerations in the transverse plane at the time of contact of the buttocks with the ground during the fall were consistent with the predictions resulting from the theoretical biomechanical assumptions in this regard.

## 2. Materials and Methods

### 2.1. Research Material

The research material is the same as described in the already published article [11]. A total of 41 physical education students aged 19–26 were qualified for the study. They were then divided into two research groups: A and B. The division of the subjects into two groups was made according to their use of a different technique of backward falls. This was the result of their practicing different sports. Group A included 19 students practicing the martial arts of aikido and ju-jitsu. In the course of their training sessions, they developed the ability to fall backwards using a technique with the side aligning of the body (Figure 2a). The students admitted that they had been instructed on the principles of performing this fall technique in their classes. Group B consisted of 22 students who were training for handball for at least four years in sports clubs of the first or second division. The students from this group during their backwards falls relied on the technique of the fall performed in a similar way to a gymnastic backward roll (Figure 2b). In this fall technique, students typically did not roll over the head in the final stage of the fall, but over the shoulder line [25]. The students in this group admitted there was no special class explaining the proper performance of this fall technique in the course of their training [10,34]. The average height of students in group A was 175 ± 4.5 cm, and their weight was 80.9 ± 7.9 kg; in group B, in turn, the height range was 181 ± 6.2 cm, and the weight was 82.1 ± 8.4 kg. There were no selection criteria imposed on the study groups as to the height and weight of students—the selection was random. The research was conducted in the period 2015–2018. Informed consent was obtained from all participants of the research. The study was conducted with the Declaration of Helsinki in mind, and the protocol itself was approved by the Commission for Bioethics at the Regional Physicians’ Council in Zielona Góra (4/55/2014).

### 2.2. Research Method

The research method described in this article resembles the procedure described in the previously published paper [11]. The research was completed at the same time, yet this article analysed the results collected from the linear acceleration motion sensor from the transverse rather than the sagittal plane. Additionally, the acceleration for the percentage time points within the 40 and 45 to 60 range, in which the buttocks of students from groups A and B contacted the ground during the fall, was analysed more thoroughly [11].

To induce backwards falls, the rotational training simulator (RTS) was used. The RTS test method validation procedure applied to diagnose posterior falls was presented in [25]. In RTS-induced falls, a person holds on to a pole while standing on a board that was accelerated to a desired speed. At the sound signal, the person let go of the pole and the board came to an abrupt halt, which, in turn, induced the person’s fall as a result of inertial forces. Investigators who observe falls at lower speeds may exclude from further research participants whose falling technique may pose a threat to their health. Head acceleration values obtained during the fall were analysed for the speed at which the board was stopped, i.e., V = 1.5 m/s. The experiment participants were involved in two tests. In the first one, they were told not to attempt to protect themselves against the fall when the inertial forces that made them fall first acted on them. This test was called the ‘immediate fall test’ (IFT). The very falling technique is sometimes adopted by sports players to reduce the risk of injury or to solicit a more favourable verdict from the referee. In the other test, the students only fell when the force causing them to fall was big enough to make them fall. The participants strove to keep their balance in an attempt to delay the fall. This test was named the ‘forced fall test’ (FFT). It can be concluded that, for FFTs, the fall imposed is consistent with the WHO definition [10,11,14], as the participants fall inadvertently. At the same time, the author of these articles considers the FFT to be more challenging, as students who attempt to delay the fall have far less time to assume the correct body position prior to the fall.

Wiva^®^ Science motion sensors (Letsense Group, Bologna, Italy) sized 40 mm + 45 mm + 20 mm and weighing 35 g were used in the study [11,35]. Wiva^®^ Science sensors are equipped with an IMU 9 axis-sensor (accelerometer, 3 axes; gyroscope, 3 axes; and magnetic sensor, 3 axes). The sample rate of the IMU was set at 100 Hz and the data were transmitted via Bluetooth to the computer in which they were then stored and processed using Biomech 2015 software. The study was concerned with the values of linear acceleration in the transverse plane. The sensor was attached to the subjects’ foreheads (Figure 1b), and the head acceleration values were analysed from the moment the subjects lost their balance, resulting in a fall (Figure 2a,b).

The entire fall sequence in both groups was divided into three time stages: 0% ≤ stage 1 < 40%, 40% ≤ stage 2 ≤ 60%, 60% < stage 3 ≤ 100%. In this article, the stages of the fall sequence are marked in the graphs and tables provided in the Section 3. In stage 1, there were the greatest differences in the performance of the fall between the groups. In group A, the trainee withdrew one leg, transferring loads to it through the foot in order to set the body to the lateral position, then rolling through the lower leg and thigh to the buttocks (Figure 2a) [10,11,21], while in group B (Figure 2b), after losing balance, the transition was made to the contact with the buttocks. During this time, the trainee should change their body segment alignment, leading to the appropriate adjustment of the angles β, γ, and ε (Figure 1) that take place during the collision with the ground [14,25]. In stage 2, in both groups, there was contact of the buttocks with the ground. Stage 3 of falling backward was the most common contact in both groups of the trainees, i.e., of the back with the ground. The analysis of the acceleration readings was halted when there was no danger of the head contact with the ground. This was usually the case when the head was parallel to the ground (Figure 2c). In handball players, there were also instances in which the movement of the head was stopped earlier due to hitting the ground with the buttocks at a large inclination angle of the torso relative to the ground. Then, the head was not parallel to the ground as a result of the transfer of high kinetic energy of the fall through the buttocks. In group A, once the parallel position of the head in relation to the ground was achieved, the movement resulting from the fall was still continued in a fashion similar to that presented in the video [21].

Participants who avoided the ‘hand error’ (supporting themselves while landing) during the fall were qualified for the study [9]. The ‘hand error’ reduced the kinetic energy of the fall as a result of the contact of other parts of the torso with the ground [14,36,37]. In the analysis of falls, the ‘hip error’ was ignored (it is permissible to bend the knee at an angle greater than 90°); this error only influenced the assessment criteria developed for falls using a technique similar to a gymnastic backwards roll, and yet it did not apply to the technique of falling with the lateral body position [10]. The participants did not make the ‘head error’ either, which is defined as tilting the head back when changing from vertical to horizontal positions, resulting in the head hitting the ground. In this way, the experiment was limited to examining the acceleration that the head was subject to as a result of the impact of the forces generated by other parts of the body when in contact with the ground during the fall, e.g., hitting the buttocks.

### 2.3. Statistical Methods

In the statistical calculations, the acceleration value in m/s^2^ from the sensor was considered, taking into account the gravitational acceleration g. This means, for example, that the given value of 0.1 acceleration means 0.1 g (0.1 × 9.81 m/s^2^ = 0.981 m/s^2^). The beginning of the measurement of linear acceleration in the transverse plane given by the sensor started from the value of 1 g. The static methods used in this article are similar to the statistical methods used for the sagittal acceleration test [11].

The times of performing the FFT and IFT exercises differed most often for the subject and between the subjects. For the number of measurements n, 100% of the time is assumed, and for the k-th measurement, the percentage time of the entire exercise is calculated using the k × 100/n% formula, where n is the number of measurements for a given individual. Acceleration was calculated for the percentage time points of the exercise from 0, 5 every 5 to 100 with interpolation. For statistical calculations, only the accelerations assigned to the percentage time points of execution from 0, 5, and 10 every 5 to 100 were used for all individuals, and the average accelerations from all individuals for IFT and FFT and in groups A and B were calculated for them. In the statistical study, basic characteristics were used, i.e., mean values, n, standard deviations, and minimum and maximum values for IFT, FFT, delta IFT, and delta FFT were calculated in each group separately. To compare the mean IFT with FFT, the Student’s *t*-test for dependent variables was used, as the IFT and FT T tests were used for the same subjects (Table 1 and Table 2). To compare groups A with B separately for IFT and FFT and delta IFT and delta FFT, the Student’s *t*-test was used for independent variables (Table 3, Table 4 and Table 6), as the means of different people are compared. This test for IFT and FFT was performed for each time point from 0 (5) 100. The last rows of Table 1, Table 2, Table 3 and Table 4 shows the means of the 21 means for IFT and FFT in groups A and B. The Student’s *t*-test for dependent variables compared the means of 21 time points for IFT with FFT in each group A and B separately and the Student’s *t*-test for independent variables between groups A and B for IFT and FFT separately. Standard deviations in groups A and B for the means differed for IFT and FFT. The quotient of the variance F test was used to test the null hypothesis; the variances in both groups were the same against the alternative, where the variances are different. This test was used for each 0 (5) 100 time point separately for IFT and FFT. The hypothesis of the equality of variance for *p* < 0.05 was rejected. The probability values < 0.05 obtained in the tables are shown in bold.

## 3. Results

In the tables, the time intervals in which the participants’ buttocks contacted the ground during a fall are marked with red letters. The individual stages of the fall are separated on the graphs by a vertical dashed line.

Figures accompanying the presented Table 1, Table 2, Table 6, Table 7 and Table 8 are listed in the Appendix A.

By analysing the charts of the linear acceleration of the head in the transverse plane versus the time of falling backwards in the form of IFT and FFT for the selected participant in groups A and B, it can be concluded that the largest positive and negative values of the head acceleration are distributed differently within the groups (Figure 3 and Figure 4). The highest values are in group A for the stage 1 fall and in group B for stage 2.

Table 1 (Appendix A) presents the dependence of the mean head acceleration values for IFT and FFT on the backward fall time in group A. Table 2 (Appendix A) shows these relationships for group B. The values of the minimum and maximum accelerations achieved in groups for IFT and FFT are presented in Table 1 and Table 2. The data show that, from the time points 40 and 45 to 60, the maximum acceleration values and the minimum values were greater in group B in absolute terms. Table 1 shows at which time points there are significant differences in the mean acceleration values between IFT and FFT in group A, while Table 2 shows the same for group B.

For most time points, the means in Table 1 are not significantly different from each other. IFT differs significantly from FFT at the 15, 20, 25, 50, and 90 time percentages (*p* < 0.05). The mean for all time points, however, did not differ significantly between IFT (0.22) and FFT (0.20) in group A. In group B (Table 2), IFT differed significantly from FFT at the time points 55 and 65 (*p* < 0.05). In group B, the means for all time points for IFT (0.40) and FFT (0.61) differed more than in group A, but the differences were not statistically significant (Table 1 and Table 2, Appendix A). Table 3 and Table 4 show the group differences for the mean acceleration values at the time points between the groups for IFT (AIFT and BIFT) and FFT (AFFT and BFFT). For group A, the mean IFF = 0.2164, and for group B, the mean IFF = 0.3960. For group A, the mean for FFF = 0.2037, and for group B, the mean FFF = 0.6126. The means differed significantly between the groups (*p* = 0.0136) for FFT, but there was no difference for IFT.

The differences between the maximum and minimum values affect the standard deviation. At the time points from 40 and 45 to 60 in group A, standard deviations were smaller than in group B (Table 1 and Table 2). In group B, there were standard deviations > 1. In Table 5, the values for delta = max (acceleration) − min (acceleration) were calculated for each person separately for IFT and FFT. There were larger delta values for group B than A. The largest delta value was for BIFT, and it also showed the highest standard deviation. Table 6 (Appendix A) shows the difference in deltas between the groups with respect to IFT and FFT. There was a significant difference between groups A and B only for IFT.

Apart from comparing IFT and FFT means between groups, it is interesting to compare measures of dispersion around these means, i.e., variance (Table 7 and Table 8, (Appendix A). In Table 7 for IFT at time points 0 to 20, the standard deviation in group A was greater than in group B, while at the other points, the standard deviation in group A was smaller than in group B. The biggest difference was at time points 40 to 60, and at these points in group B, there was a large standard deviation. In Table 8 for FFT at time points 0 to 15, the standard deviation in group A is greater than in group B, while at the other points, the standard deviation in group A is smaller than in group B. The biggest difference is at the time points 55 to 65, and at these points in group B, a large standard deviation was obtained.

## 4. Discussion

The results of the research obtained using the RTS rotary trainer in forcing the fall backwards revealed that the susceptibility to head injuries depends on the learnt fall technique used by the students. The applied method of assessing the susceptibility [10,14] to body injuries revealed that students training handball and specific martial arts acquired appropriate motor habits to protect their head from hitting the ground during a fall. Students not practicing a specific sports discipline, whose physical fitness was mainly based on the physical education program implemented as part of school education, had a much greater susceptibility to head injuries.

However, despite the fact that the head did not hit the ground during the backward fall, the very impact of the buttocks on the ground at a specific positioning of body segments may transmit high acceleration through the biokinematic chain which may, in turn, adversely affect the head. To demonstrate this, it was necessary to use appropriate sensors [11]. It is difficult to analyse the forces at work in head acceleration during a fall in detail as they result from forces generated by the muscles, forces generated by the contact of body segments with the ground during the fall, and forces generated by the curvilinear motion within the gravitational field. Thus far, no detailed biomechanical studies on this topic have been performed. The biomechanical analysis of the forces acting on the head and resulting from the forces generated by hitting the buttocks against the ground was developed by Mroczkowski [14].

The analysis of the frames of the film recorded during the fall showed that when the buttocks hit the ground, there may be a head movement that tilts it backwards and also forwards. The biomechanical analysis of the forces acting when the buttocks hit the ground predicted that the head could perform such a movement during a fall due to the resultant moment of force acting on it. These effects occurred mainly within the group of students who fell using a fall technique similar to a gymnastic backward roll. Such information was confirmed by analysing the accelerations obtained by the sensor mounted on the head [11]. The information provided by the sensor is important as, on the basis of such data, it is possible to obtain information about the risk of head injury in the case of a fall in real, non-laboratory conditions, which is provided by a normal ground, such as turf, pitch, etc.; in these cases, the acceleration at work will be much greater. The greatest acceleration values were generated on the head in group B when the buttocks hit the ground during falls. The information provided on the acceleration of the head throughout the fall may offer information as to the susceptibility to damage of other parts of the body, in particular, the pelvis and cervical spine linked in the biokinematic chain [11].

From the biomechanical analysis presented in Figure 1, the results showed that the increased susceptibility to damage to the head and the cervical spine is informed by the values of acceleration in the sagittal and transverse planes. The information on the change of the sense of the acceleration vectors, which can be determined on the basis of the change in the sign of the acceleration values, is also important. This was confirmed, for example, by the movement of the head during the backward fall, as well as forward when analysing the acceleration in the sagittal plane. Due to the complexity of the forces at work during a fall, the analysis of the resultant value of these accelerations given by the sensor seems less important, and the direction and sense of the resultant vector is more important. The head, naturally, should not be accelerated rapidly during a fall, so as to not generate large inertia forces. The results of research on acceleration in the sagittal plane for the same research group were published [11]. Greater differences were found in the adopted values of the minimum and maximum acceleration in group B for both IFT and FFT. The difference was between the maximum and minimum acceleration of the so-called deltas. The value of these deltas was much higher for people from group B compared to group A. This article examined the head acceleration values in the transverse plane. The value of these deltas for these accelerations was definitely higher for people from group B compared to group A (Table 5). The time points from 40 and 45 to 60, in which the buttocks came into contact with the ground during a fall, were subjected to a special analysis. The biomechanical analysis showed that accelerations occurring when the buttocks hit the ground were generated in this interval. Greater differences were found in the adopted values of the minimum and maximum acceleration in group B for both IFT and FFT in Table 1 and Table 2 for time points 40 and 45 to 60. The above results suggest that the fall technique used in group A resulted in motor habits that did not generate large changes in the values of the obtained head accelerations in the transverse plane at the moment of contact with the buttocks with the ground compared to group B, in particular, for the time period when the buttocks contacted the ground.

The article analysed the results in terms of the transfer of motor habits between the IFT and FFT tests in groups and between groups, as performed for the transverse plane. Movement habits were transferred in group A between IFT and FFT tests, which is evidenced by the lack of statistical differences between the mean acceleration values for the entire group for these tests and a small standard deviation (Table 1). It can be concluded that movement habits were significantly worse in group B between IFT and FFT tests due to a greater standard deviation in a much greater number of time points than in group B, despite the lack of significant statistical differences between the mean values of acceleration (Table 2). Table 4 shows that there are significant differences between the mean values of acceleration in groups A and B in the performance of the FFT test, which attests to different movement habits in these groups. However, no such significant differences were found between the groups in the IFT test (Table 3).

The above-described acceleration results for the transverse plane show smaller statistical differences between the performance of the IFT test and the FFT test in groups and between groups compared to the accelerations obtained for the sagittal plane [11]. However, the smaller standard deviations obtained in group A indicate a better transfer of motor habits in this group compared to group B. The obtained differences seem to be justified in the results for acceleration in the sagittal plane and in the transverse plane. The acceleration achieved by the head in the sagittal plane is mainly responsible for the backward or forward motion of the head when falling backward. Head movement in the transverse plane is restricted due to anatomical limitations. This may affect the statistical differences obtained between the IFT and FFT tests in the groups and between the groups.

The results are interesting in terms of the transfer of motor habits in the time interval from 40 and 45 to 60, in which the buttocks came into contact with the ground during a fall. For the IFT test, no significant differences were found between the groups for the mean values at time points from 40 and 45 to 60 (Table 3); however, the standard deviation in group B at these points was significantly higher than in group A (Table 7). For the FFT test, no significant differences were found between the groups for the mean values at the time points 40 and 60, but they were found for 45, 50, and 55 (Table 4). Standard deviations in group B at time points from 40 and 45 to 60 were significantly higher than in group A (Table 8). The above results suggest that the fall technique applied in group A gives motor habits that do not generate large changes in the value of the obtained head acceleration at the moment of contact of the buttocks with the ground compared to group B.

It should be noted that, for the IFT at time points from 0 to 20, the standard deviations in group A were larger than in group B (Table 7), and for the FFT at time points from 0 to 15 (Table 8). At the same time, the positive and negative values of the average acceleration of the group (Table 1 and Table 2) and the selected student (Figure 3 and Figure 4) at the beginning of the first stage of the fall were higher in group A compared to group B.

Inverse relationships between the groups were observed in stage 2 of the fall. The explanation of these observations may be offered by way of performing the falls in the groups (Figure 2a,b). Obtaining greater accelerations in group A in the first stage of the fall may lead to transferring ground reactions to the head through the trunk and lower limb as a result of rolling on it. This reaction is especially visible when the lower limb is withdrawn and the entire load is transferred through the foot at the beginning of this stage [21]. Performing these motor activities requires the acquisition of certain motor habits. The analysis of this phase of the fall requires extending the research for the fall technique used in group A. In group B, the fall technique in this phase resembles the behaviour of a vertical pole once it has fallen. One can imagine there is a head at the top of the pole. It is obvious from the mechanical perspective that the acceleration of the head in the transverse plane indicated by the sensor will be less than 1 g. This may account for the assumption of lower head acceleration values in stage 1 of the fall in group B.

For a full analysis of the acceleration acting on the head during the contact of the buttocks with the ground, the component of acceleration in the frontal plane should be taken into account, which may constitute the subject of the next publication. However, preliminary analyses showed that there were no differences between the studied groups in this phase of the fall. It would be interesting to analyse the acceleration in the frontal plane to estimate its value at the moment of stopping the board on which the participant is standing, just before the fall, as in the first phase. It would be interesting in terms of biomechanical interpretation.

In general, it should be noted that the results collected about the transfer of motor habits in the performance of the IFT and FFT test in groups and between groups were similar for both the results obtained for the acceleration present in the transverse and sagittal plane. Conclusions similar to those obtained from the analysis of accelerations in the sagittal plane can be drawn. It was found that physical education students using the technique of a lateral fall during a horizontal fall backward transferred better movement habits compared to students training handball in the form where the person does not resist the fall (IFT) and the form where they do not crash (FFT).

The obtained results of linear acceleration in the transverse and sagittal planes [11] do not allow for an unequivocal statement that the fall technique similar to the gymnastic backward rollover is incorrect when performing a fall using RTS. The worse results of handball players may have been caused by the fact that they had not yet undertaken special training regarding the principles of falling backwards safely [11,34]. Possibly, other sport groups failing this technique would have achieved better results.

The selection of the correct fall technique should depend on the direction of the forces at work [11]. For athletes, there is no perfect technique, as it depends on the discipline [25,38,39]. Falling backwards using a technique similar to the gymnastic backflip is justified when the resultant force causing the fall depends mainly on the vertical component. This occurs mainly in players practicing disciplines where jumps in the vertical direction are frequent, i.e., volleyball, basketball, handball [25]. After completing education and training, a large part of the population no longer practices such sports. A frequent cause of falling is a slip, which is dominated by the horizontal component of the force inducing the fall [25,40]. It is a frequent occurrence in everyday physical activity, especially in the elderly. During everyday movement, practicing vertical jumps is rare, especially in the elderly.

A horizontal force was used to induce a fall on the RTS [10,11,25]. The obtained results suggest that the appropriate technique for such cases is the fall technique with a lateral body position. It seems reasonable that fall technique training should not teach one to roll over their head. As research using the RTS showed, in a group of approximately 800 physical education students, no appropriate movement of the upper limbs was found when falling backwards, which could protect the head in the event of rolling over it [10,11]. It follows from the above considerations that, since the horizontal force is a frequent cause of falls, teaching the technique of falling backwards with a lateral body position should be included in the school curriculum [11]. At the same time, in order for the movement habits to consolidate, it is recommended to practice this fall technique throughout one’s life as part of one’s own physical safety. It also seems reasonable based on reports by other authors to state that the use of the lateral positioning technique may prevent hip fractures during a fall [41,42,43].

The RTS allows for the diagnosis of movement habits during falls for various groups of athletes to determine the usefulness of a specific technique of a fall induced by a force with a horizontal direction [10,11,25]. It is difficult to find similar design solutions that give the respondents the appropriate speed in rotational motion, forcing a fall by using the force of horizontal inertia resulting from a sudden stop in the movement that the examined person is performing. Therefore, it is difficult to compare the obtained results with the results of other authors. Work is underway to popularize the use of this device. The advantage of the device used to test backward falls compared to non-instrumental methods is that the fall is forced by an external force [10,11,37]. However, one should be aware of the limited applicability of the RTS for adults with a high level of physical fitness. The device is intended mainly for people for whom a fall is a frequent event in connection with the sports discipline or work performed, e.g., in uniformed services. The forces with which an RTS can induce a fall may be too much for people who do not have proper motor habits during a fall, particularly the elderly [11].

The results presented in this article diagnose the susceptibility to head, neck, and pelvic injuries during a fall. It would be beneficial to increase the number of sensors attached not only to the head, especially when determining the susceptibility to pelvic damage. Attempts were made in this regard, but none of the results obtained were as accurate as those obtained from the sensor on the head due to the problem of attaching the sensors to other body parts. The sensor on the head was pressed against the forehead with a band that adequately limited the possibility of its movement in the course of the tests.

## 5. Conclusions

The research compared the backward-forced fall techniques of students using the immediate fall test (IFT) and the forced fall test (FFT). The training simulator (RTS), which is mainly intended for examining adults who often fall as a result of their sports discipline or work, was used to induce backward falls. The study compared the backward fall techniques in the representatives of various sport disciplines. The groups included students practicing martial arts that performed falls using the technique with the side aligning of the body and handball players that performed falls using the technique similar to a gymnastic backward roll. The conclusions obtained from the results of linear head acceleration in the transverse plane can be considered similar to the conclusions obtained earlier for the same research groups for the acceleration in the sagittal plane. It was found that, in physical education, for students using the technique of falling with a lateral position of the body during a backward fall caused by a horizontal force, movement habits are transferred better compared to students training handball in a situation when the person does not resist falling (IFT) and the form when not going to fall (FFT). The greatest differences in the acceleration in the transverse plane occurred between these groups during the contact of the buttocks with the ground during a backward fall, which was consistent with the previous biomechanical assumptions. The lower changes in head acceleration obtained in physical education students falling with a lateral body position than in students training handball indicate their lower susceptibility to head, cervical spine, and pelvis injuries when falling backwards caused by horizontal force. The obtained results confirm the validity of the recommendation to include the technique of falling with a lateral body position in school education due to the fact that horizontal force is a frequent cause of falls.

## Figures and Tables

**Figure 1 sensors-23-03276-f001:**
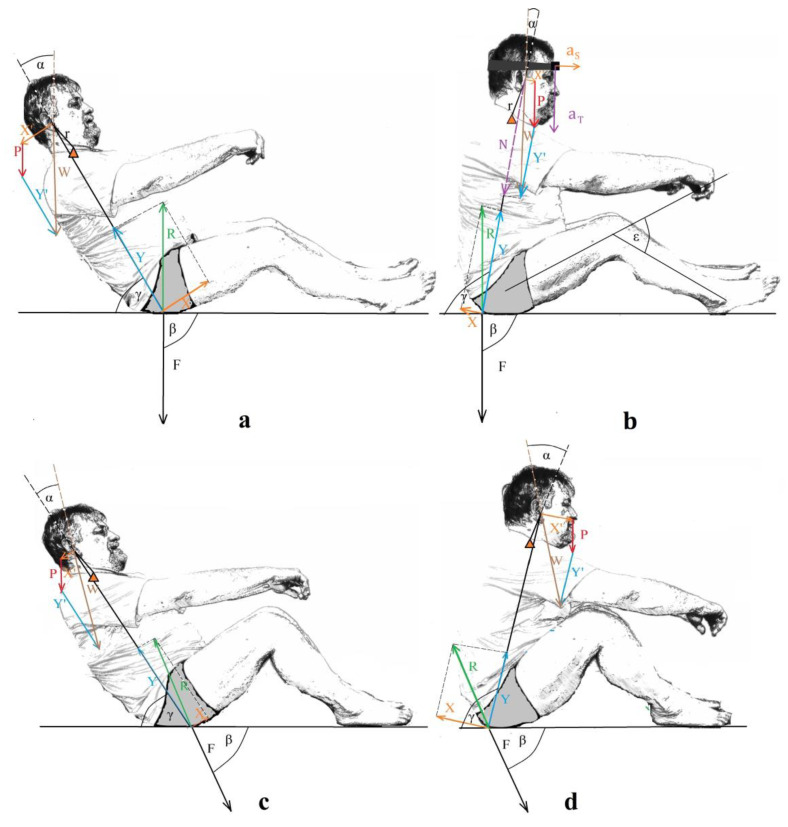
Relationship between the forces acting when the buttocks hit the ground and the accelerations indicated by the sensor placed on the participant’s head. (ß, angle of inclination between the direction of the force **F** and the horizontal plane of the fall; γ, angle between the torso and the horizontal plane; ε, knee bend angle. The (**a**,**b**) show the fall with identical values of the angle ß and varying values of the angle of the plane γ—as do the (**c**,**d**).

**Figure 2 sensors-23-03276-f002:**
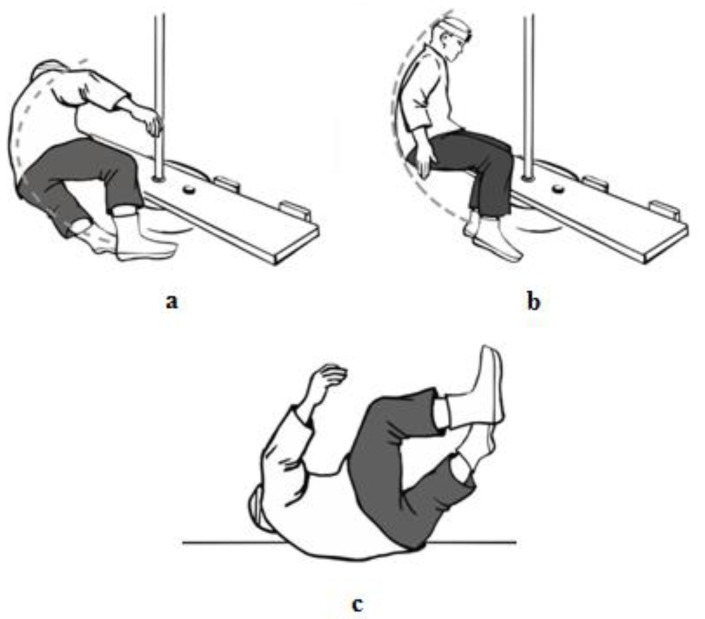
Types of backward fall techniques as adopted by the participants, and the presentation of the moment of completion of the head acceleration reading during the tests. (**a**) Fall performed backward with the side aligning of the body, (**b**) fall performed in a way similar to a gymnastic backward roll, (**c**) moment of completion of the head acceleration reading.

**Figure 3 sensors-23-03276-f003:**
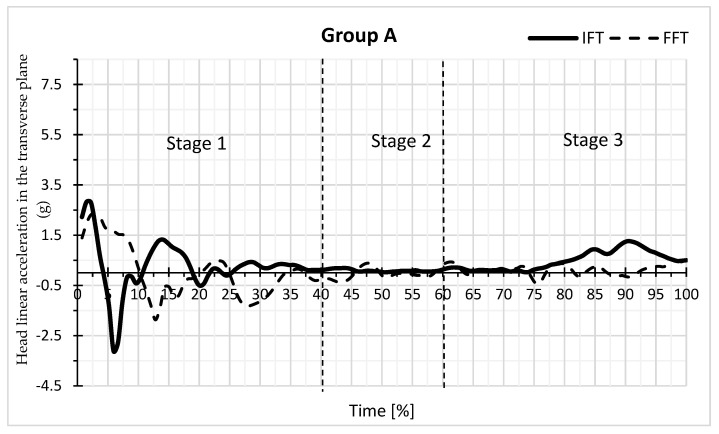
The dependence of the value of linear head acceleration in the transverse plane on the time of the backwards fall of the IFT and FFT type in group A for a selected participant.

**Figure 4 sensors-23-03276-f004:**
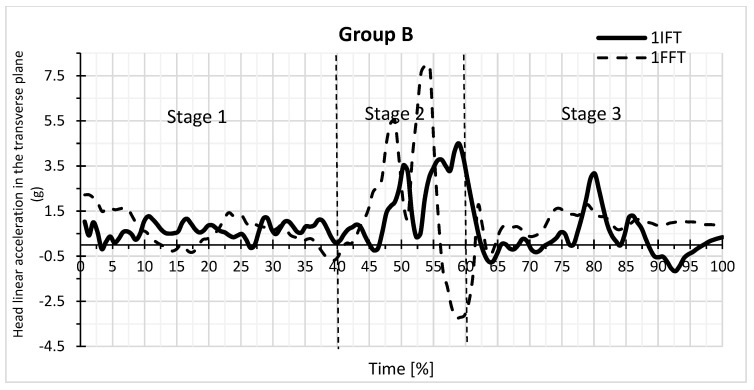
The dependence of the value of linear head acceleration in the transverse plane on the time of the backwards fall of the IFT and FFT type in group B for a selected participant.

**Table 1 sensors-23-03276-t001:** Basic characteristics (mean, standard deviation SD, max, and min) for IFT and FFT for group A. For each time point, the differences between IFT and FFT and Student’s *t*-tests were calculated for dependent variables; it was checked whether this was significantly different from zero.

Time %	Mean A IFT	SD A IFT	Min A IFT	Max A IFT	Mean A FFT	SD A FFT	Min A FFT	Max A FFT	Difference (IFT–FFT)	*p*
0	1.84	1.15	−0.58	3.58	2.56	1.49	−0.42	5.03	−0.71	**0.0128**
5	0.07	1.84	−1.62	3.53	−0.58	2.23	−3.51	2.38	0.64	0.2265
10	−0.23	1.69	−1.85	4.37	−0.01	0.98	−2.61	1.52	−0.22	0.5420
15	0.26	1.57	−3.76	1.54	−0.53	1.27	−2.86	1.25	0,80	**0.0292**
20	−0.99	0.97	−2.38	0.46	0.24	0.77	−1.44	1.51	−1.24	**0.0001**
25	−0.14	0.40	−0.89	0.39	0.43	0.99	−0.65	2.43	−0.56	**0.0373**
30	0.16	0.47	−1.02	0.72	−0.13	0.85	−1.32	1.35	0.29	0.1709
35	0.15	0.25	−0.48	0.46	−0.11	0.63	−1.22	1.26	0.26	0.1657
40	0.17	0.20	−0.27	0.50	0.04	0.36	−0.80	0.62	0.12	0.2011
45	0.20	0.16	−0.08	0.58	−0.03	0.45	−0.79	0.61	0.23	0.0559
50	0.20	0.25	−0.45	0.75	−0.08	0.48	−1.29	0.35	0.27	**0.0128**
55	0.10	0.23	−0.50	0.39	0.13	0.29	−0.28	0.90	−0.03	0.8068
60	0.11	0.23	−0.40	0.41	0.22	0.18	−0.11	0.51	−0.11	0.1641
65	0.03	0.31	−0.72	0.42	0.13	0.19	−0.23	0.42	−0.10	0.1540
70	−0.11	0.61	−1.76	0.39	0.02	0.40	−1.07	0.37	−0.13	0.3638
75	0.08	0.23	−0.53	0.41	0.01	0.63	−1.74	0.55	0.08	0.6251
80	0.32	0.16	−0.01	0.59	0.20	0.68	−1.42	1,12	0.12	0.4549
85	0.49	0.45	−0.41	1.08	0.44	0.45	−0.38	1.14	0.05	0.6502
90	0.70	0.41	0.07	1.38	0.42	0.40	−0.15	1.05	0.28	**0.0314**
95	0.56	0.40	−0.23	1.02	0.42	0.33	−0.12	0.95	0.13	0.2710
100	0.58	0.33	−0.18	0.95	0.48	0.32	−0.02	1.12	0.09	0.4270
Mean	0.22	0.51			0.20	0.61			0.01	0.8966

**Table 2 sensors-23-03276-t002:** Basic characteristics (mean, standard deviation SD, max, and min) for IFT and FFT for group B. For each time point, the differences between IFT and FFT and Student’s *t*-tests were calculated for dependent variables; it was checked whether this was significantly different from zero.

Time %	Mean B IFT	SD B IFT	Min B IFT	Max B IFT	Mean B FFT	SD B FFT	Min B FFT	Max B FFT	Difference (IFT–FFT)	*p*
0	0.81	0.66	−0.77	1l70	0.76	0.77	−0.62	2.22	0.05	0.7929
5	0.41	0.72	−0.85	2.14	0.69	0.78	−0,72	2.08	−0.28	0.0957
10	0.64	0.62	−0.51	1.96	0.60	0.77	−0.38	2.50	0.04	0.8274
15	0.88	0.58	−0.38	2.23	0.88	1.19	−0.63	4.69	−0.01	0.9862
20	0.73	0.74	−0.56	2.94	1,04	1.16	−2.03	3.05	−0.31	0.2395
25	0.81	1.23	−2.62	3.98	0.88	1.34	−3.34	2.99	−0.07	0.8784
30	1.47	1.68	−1.23	5.45	0.73	0.92	−1.17	2.77	0.74	0.0780
35	0.70	1.58	−3.52	3.35	0.72	0.86	−0.60	3.40	−0.03	0.9446
40	0.98	2.46	−3.26	7.88	0.44	1.24	−2.85	2.23	0.53	0.2951
45	0.01	2.23	−6.64	3.46	0.77	0.93	−1.26	2.58	−0.75	0.1328
50	0.30	2.46	−6.68	7.03	1.10	1.35	−1.68	4.53	−0.80	0.2544
55	−0.14	1.71	−4.24	3.42	1.38	2.14	−1.42	7.97	−1.52	**0.0188**
60	0.41	1.84	−3.56	3.34	0.25	1.96	−2.99	3.94	0.16	0.8084
65	−0.34	1.85	−5.28	2.80	1.19	1.38	−0.82	4.10	−1.53	**0.0016**
70	−0.22	1.56	−4.70	2.27	−0.11	1.58	−3.08	3.28	−0.10	0.7186
75	0.27	0.88	−1.03	2.82	0.28	1.35	−1.81	2.79	−0.01	0.9833
80	0.11	1.16	−3.25	3.12	0.24	1.80	−3.65	4.40	−0.14	0.7469
85	0.07	1.34	−5.10	2.16	0.19	0.89	−2.13	1.51	−0.12	0.5783
90	0.07	0.70	−1.61	1.19	0.20	1.10	−3.75	1.65	−0.13	0.4459
95	0.35	1.18	−1.31	4.56	0.25	0.71	−1.03	1.84	0.10	0.6135
100	0.05	1.03	−3.82	0.97	0.36	0.73	−1.31	1.32	−0.31	0.1091
Mean	0.40	0.46			0.61	0.39			0.55	0.0881

**Table 3 sensors-23-03276-t003:** Comparison of the transverse linear acceleration of the head with Student’s *t*-tests for independent variables for IFT between groups A and B.

Time %	Mean A IFT	Mean B IFT	Difference	*p*
0	1.8426	0.8114	1.0313	**0.0009**
5	0.0682	0.4094	−0.3412	0.4271
10	−0.2316	0.6361	−0.8677	**0.0308**
15	0.2604	0.8781	−0.6176	0.0944
20	−0.9944	0.7308	−1.7252	**0.0000**
25	−0.1351	0.8116	−0.9467	**0.0026**
30	0.1621	1.4730	−1.3109	**0.0022**
35	0.1547	0.6952	−0.5405	0.1479
40	0.1653	0.9770	−0.8117	0.1596
45	0.1966	0.0125	0.1842	0.7216
50	0.1955	0.3011	−0.1056	0.8535
55	0.1022	−0.1390	0.2412	0.5458
60	0.1122	0.4080	−0.2958	0.4922
65	0.0268	−0.3409	0.3677	0.3992
70	−0.1122	−0.2156	0.1034	0.7877
75	0.0813	0.2723	−0.1910	0.3631
80	0.3203	0.1052	0.2151	0.4264
85	0.4946	0.0658	0.4289	0.1898
90	0.7024	0.0680	0.6344	0.0014
95	0.5567	0.3522	0.2045	0.4751
100	0.5753	0.0464	0.5289	**0.0385**
Mean	0.2164	0.3960	−0.1796	0.2308

**Table 4 sensors-23-03276-t004:** Comparison of the transverse linear acceleration of the head with Student’s *t*-tests for independent variables for FFT between groups A and B.

Time %	Mean A FFT	Mean B FFT	Difference	*p*
0	2.5558	0.7609	1.7949	**0.0000**
5	−0.5765	0.6920	−1.2685	**0.0167**
10	−0.0111	0.5960	−0.6070	**0.0321**
15	−0.5347	0.8835	−1.4183	**0.0007**
20	0.2435	1.0393	−0.7958	**0.0149**
25	0.4287	0.8820	−0.4534	0.2325
30	−0.1314	0.7322	−0.8636	**0.0035**
35	−0.1052	0.7204	−0.8255	**0.0013**
40	0.0417	0.4442	−0.4025	0.1801
45	−0.0322	0.7669	−0.7990	**0.0015**
50	−0.0787	1.1034	−1.1821	**0.0008**
55	0.1283	1.3847	−1.2564	**0.0153**
60	0.2244	0.2512	−0.0268	0.9531
65	0.1280	1.1941	−1.0661	**0.0019**
70	0.0198	−0.1125	0.1323	0.7244
75	0.0053	0.2791	−0.2738	0.4217
80	0.2029	0.2437	−0.0408	0.9263
85	0.4447	0.1886	0.2561	0.2643
90	0.4191	0.2021	0.2170	0.4224
95	0.4230	0.2530	0.1700	0.3426
100	0.4816	0.3591	0.1225	0.5035
Mean	0.2037	0.6126	−0.4089	**0.0136**

**Table 5 sensors-23-03276-t005:** Basic characteristics (number of observations N, mean, standard deviation SD, min, and max) for the delta variable = max (accelerations) − min (accelerations).

Variable	N	Mean	Minimum	Maximum	Std. Deviat.
AIFT	19	4.2476	1.7160	7.2350	1.6715
AFFT	19	5.2437	2.8550	8.3880	1.5784
BIFT	22	6.8337	3.9620	14.5560	2.2266
BFFT	22	5.6100	1.8380	10.4770	2.1339

**Table 6 sensors-23-03276-t006:** Comparison of independent variables, and mean delta values for IFT and FFT between groups A and B with Student’s *t*-tests.

	Mean Group A	Mean Group B	*p*
AIFT vs. BIFT	4.2476	6.8337	**0.0002**
AFFT vs. BFFT	5.2437	5.6100	0.5413

**Table 7 sensors-23-03276-t007:** Standard deviations for IFT in groups A and B as well as the quotient of variance F test and probability *p* for testing the equality of variance in groups A and B.

	Std. Deviat.	Std. Deviat.	Quotient F	*p*
Time %	AIFT	BIFT	variance	variance
cz0	1.1459	0.6558	3.0536	**0.0159**
cz5	1.8418	0.7164	6.6098	**0.0001**
cz10	1.6933	0.6177	7.5140	**0.0000**
cz15	1.5709	0.5848	7.2158	**0.0000**
cz20	0.9685	0.7437	1.6957	0.2458
cz25	0.3971	1.2252	9.5175	**0.0000**
cz30	0.4739	1.6814	12.5879	**0.0000**
cz35	0.2531	1.5757	38.7698	**0.0000**
cz40	0.1953	2.4569	158.2053	**0.0000**
cz45	0.1597	2.2279	194.5936	**0.0000**
cz50	0.2528	2.4603	94.7015	**0.0000**
cz55	0.2284	1.7091	55.9947	**0.0000**
cz60	0.2317	1.8438	63.3288	**0.0000**
cz65	0.3124	1.8545	35.2321	**0.0000**
cz70	0.6079	1.5616	6.6003	**0.0002**
cz75	0.2350	0.8763	13.9080	**0.0000**
cz80	0.1571	1.1556	54.0776	**0.0000**
cz85	0.4473	1.3357	8.9172	**0.0000**
cz90	0.4142	0.7036	2.8855	**0.0269**
cz95	0.4039	1.1760	8.4791	**0.0000**
cz100	0.3335	1.0292	9.5211	**0.0000**

**Table 8 sensors-23-03276-t008:** Standard deviations for FFT in groups A and B as well as the quotient of variance F test and probability *p* for testing the equality of variance in groups A and B.

	Std. Deviat.	Std. Deviat.	Quotient F	*p*
Time %	AFFT	BFFT	variance	variance
cz0	1.4915	0.7739	3.7145	**0.0049**
cz5	2.2295	0.7826	8.1167	**0.0000**
cz10	0.9816	0.7651	1.6461	0.2730
cz15	1.2698	1.1949	1.1294	0.7821
cz20	0.7711	1.1570	2.2517	0.0869
cz25	0.9916	1,3428	1.8338	0.1982
cz30	0.8472	0.9174	1.1726	0.7388
cz35	0,6299	0.8577	1.8540	0.1904
cz40	0.3636	1.2383	11.5963	**0.0000**
cz45	0.4538	0.9303	4.2037	**0.0033**
cz50	0.4776	1.3489	7.9765	**0.0000**
cz55	0.2918	2.1390	53.7226	**0.0000**
cz60	0.1795	1.9614	119.4337	**0.0000**
cz65	0.1918	1.3825	51.9642	**0.0000**
cz70	0.3981	1.5791	15.7371	**0.0000**
cz75	0.6285	1.3471	4.5937	**0.0019**
cz80	0.6814	1.7982	6.9631	**0.0001**
cz85	0.4531	0.8900	3.8584	**0.0054**
cz90	0.3992	1.1040	7.6487	**0.0001**
cz95	0.3287	0.7072	4.6283	**0.0018**
cz100	0.3189	0.7319	5.2660	**0.0008**

## Data Availability

The datasets analysed during the current study are available from the corresponding author on reasonable request.

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
