# Peer review of "Influence of the Backward Fall Technique on the Transverse Linear Acceleration of the Head during the Fall"

_sensors, 2023, doi:10.3390/s23063276_

Round 1

Reviewer 1 Report

Kindly rewrite the method section (Line 16- Line 19). None of the group is mentioned in the conclusion section. which group is prone to injuries etc.

The originality and novelty of the study need to be highlighted, and the gap of the study is still unclear. 

Line 117-125: Kindly rewrite this paragraph as it will make the reader confused about the aim of the study.

Remove the term "(Fig. 1)" Line 103, from introduction section and move it to the material and method section (Line 179-198). The figures (1, 2, 3 and 4) can be combined as a single figure and explanation should be provided in the methodology section.

How did the authors calculate the sample size?

Instead of drawing yellow line it's better to highlight the contact in another alternative way rather than drawing line as it disturbs the average line graph. 

Kindly choose only one either table or figure as the information is almost similar. The important p values you can mention in the results section. In my opinion the rest of the results can be provided as supplementary document. 

  Kindly provide limitation of the study and possible future studies. 

Author Response

Response to Reviewer 1  Comments

All changes made to the text (manuskrypt) recommended by the reviewers are highlighted in yellow. In the article all changed sentences are marked in yellow color.

 Kindly rewrite the method section (Line 16- Line 19).

Changes have been made to the lines 17 -20 None of the group is mentioned in the conclusion section. which group is prone to injuries etc.

Changes have been made to the lines 17 -20

,,Group A consisted of 19 martial arts practitioners who, during the study, performed falls using the side aligning of the body technique. Group B consisted of 22 handball players who, during the study, performed falls using the technique performed in a way similar to a gymnastic backward roll.”

Changes have been made to the lines 583 - 587

,,The study compared the backward fall techniques in the representatives of various sports disciplines. The groups included students practicing martial arts, performing falls using the technique with side aligning of the body, and handball players performing falls using the technique performed in a way similar to a gymnastic backward roll. »

This was also explained in the lines 594 -600

The greatest differences in the acceleration in the transverse plane occur between these groups during the contact of the buttocks with the ground during a backward fall, which was consistent with the previous biomechanical assumptions. The lower changes in head acceleration obtained in physical education students falling with a lateral body position than in students training handball indicate their lower susceptibility to head injuries, cervical spine and also the pelvis, for falling backwards caused by horizontal force.

The originality and novelty of the study need to be highlighted, and the gap of the study is still unclear. 

This was also explained in the lines 510 - 514

For a full analysis of the accelerations acting on the head during the contact of the buttocks with the ground, the component of acceleration in the frontal plane should be taken into account, which may constitute the subject of the next publication. However, preliminary analyses show that there are no differences between the studied groups in this phase of the fall.

Additions have been added in lines 563 – 578

The advantage of the device used to test backward falls compared to non-instrumental methods is that the fall is forced by an external force [10,11,37]. However, one should be aware of the limited applicability of the RTS for adults with a high level of physical fitness. The device is intended mainly for people for whom a fall is a frequent event in connection with the sports discipline or work performed, e.g. in uniformed services. The forces with which an RTS can induce a fall may be too much for people who do not have proper motor habits during a fall, particularly the elderly[11]. The results presented in this article diagnose the susceptibility to head, neck and pelvic injuries during a fall. It would be beneficial to increase the number of sensors attached not only to the head, especially when determining the susceptibility to pelvic damage. Attempts were made in this regard, but none of the results obtained were as accurate as those obtained from the sensor on the head, due to the problem of attaching the sensors to other body parts. The sensor on the head was pressed against the forehead with a band that adequately limited the possibility of its movement in the course of the tests.

This was also explained in the lines 140- 144

Simultaneously, it should be noted that the discussed accelerations indicated by the sensor may be affected by the participation of the head in the curvilinear movement of gravitational fields. It is difficult to find scientific studies on the subject of falling backwards. Moreover, the accelerations achieved may be influenced by the muscle force generated by the players and acting on the head during the fall.

Line 117-125: Kindly rewrite this paragraph as it will make the reader confused about the aim of the study.

Extensive changes have been made here.  The editing was accopanied by addition to the background and the introduction of an extended biomechanical description with an explanation of new fig 1. The type of changes was suggested by the second reviewer.

Changes have been made to the lines  107 -161

In order to understand it better, it is necessary to extend the considerations of Mroczkowski, who presented the biomechanical analysis of the moment of force acting on the head during a backward fall [14]. Based on his assumptions, this article presents an analysis of the relationship between the indications of the sensor mounted on the head of the falling person (Fig 1b), in terms of the direction and sense of the linear acceleration vectors in the transverse and sagittal planes, with these features of the force vectors acting on the head. Using the F = ma relation resulting from Newton's second law of motion, it is obvious that the direction and sense of the force and acceleration vectors are the same with the values being directly proportional. At the same time, the sensor, due to its stable attachment and negligible mass, can be considered as one solid body with the head.

Mroczkowski concluded that the torso and head may be considered as two links of the kinematic chain, recognizing the connection between them as articulated (Fig. 1), which in the event of a moment of force causes the head to move [14]. For this reason, it is assumed that there is a vector P corresponding to the weight of the head. If all the bodies were treated as solid during the collision, the considerations would assume that the W and R vectors should differ only in their directions. At the same time, by projecting the vector W onto the line connecting the designated point of contact of the buttocks with the ground and the centre of gravity of the head, the N vector (Fig 1b) is obtained (marked with a dashed line, as an exception). The N vector should also be equal to the resultant vector that would be created from the addition of vectors P and Y'. The N vector should only differ from the Y vector in sense. The same relationships will occur if the sensor indications were analysed in the other settings during the fall (Fig 1a,c,d). Some discrepancies from the relationships described above may result from the fact that the value of the P vector was adopted without the precise determination of the head mass. Its length may therefore be inconsistent with reality. In reality, the change of the other vectors marked in the centre of gravity of the head may be affected by the fact that not all the energy during the impact with the ground will be transferred to the head [14].The above analysis suggests that vector X' will be closest to the direction of linear acceleration in the sagittal plane aS, and vector N for the transverse plane aT. This analysis shows that the indications of aT acceleration will be the most informative in terms of the risk of the head, neck, spine and then pelvic injury as a result of the connection with the N force. On the other hand, the as indications about the risk of hitting the head during a fall due to the connection with the X' force. Simultaneously, it should be noted that the discussed accelerations indicated by the sensor may be affected by the participation of the head in the curvilinear movement of gravitational fields. It is difficult to find scientific studies on the subject of falling backwards. Moreover, the accelerations achieved may be influenced by the muscle force generated by the players and acting on the head during the fall.

The biomechanical analysis presented in Figure 1 shows that when the buttocks hit the ground, the main component of the force acting on the pelvis during the injury is pointed upwards (Y), with the smaller component (X) directed forwards or backwards. In the Young-Burgess Classification of pelvic fractures [28,29] it is the VS (Vertical Shear) type. With this type of fracture, the pubic and ischial bones limiting the obturator foramen may be fractured, which carries the risk of damage to the vascular and nervous structures passing through this hole and damage to the lesser pelvic organs.

     The age group particularly at risk of fractures, including pelvic fractures, are the elderly, who suffer from a decrease in bone mineral density and weakening of the mechanical strength of the bones associated with osteoporosis. Pelvic fractures due to a backward fall can be classified as a low-energy fractures, particularly characteristic of the elderly population. According to scientific reports, regardless of the classification into fall types, falls among people over 65 years of age are the most common cause of such fractures [30,31]. There is an increase in the incidence of low-energy pelvic fractures in people with osteoporosis [30,32]. Osteoporotic pelvic fractures in the elderly result in an annual mortality rate of 9.5% to 27% among the elderly[33].

This was also explained in the lines 170-180

The aim of the research undertaken in this article was to investigate changes in the linear acceleration of the head in the transverse plane during a backward fall forced on the rotating trainer among representatives of various sports disciplines. Those were students of physical education practicing martial arts, performing falls with the side aligning of the body technique and handball players performing falls applying the technique similar to a gymnastic backward roll. At the same time, the comparison of these results with those obtained previously in the sagittal plane was attempted. The aim of the research was furthermore to check whether the values of the obtained accelerations in the transverse plane at the time of contact of the buttocks with the ground during the fall were consistent with the predictions resulting from the theoretical biomechanical assumptions in this regard.

  1. Remove the term "(Fig. 1)" Line 103, from introduction section and move it to the material and method section (Line 179-198).

Changes have been made here. Fig 1 has been incorporated into the new Fig 1 in part b - lines 102 – 106

Figure 1. Relationship between the forces acting when the buttocks hit the ground and the accelerations indicated by the sensor placed on the participant's head. (ß- angle of inclination between the direction of the force F and the horizontal plane of the fall, γ- angle between the torso and the horizontal plane, ε – knee bend angle.

The figures (1, 2, 3 and 4) can be combined as a single figure and explanation should be provided in the methodology section.

Changes have been made to the lines 278 – 282. 

Figure 2. Types of backward fall techniques as adopted by the participants and presentation of the moment of completion of the head acceleration reading during the tests. 2a - fall performed backward with side aligning of the body, 2b - fall performed in a way similar to a gymnastic backward roll, 2c - moment of completion of the head acceleration reading

How did the authors calculate the sample size?

The answer is below

"The independent variable - X-axis is time and the dependent variable - Y-axis is acceleration. Time is presented as a percentage from 0% to 100%, as each participants had a different number of measurements n. The last measurement (nth) is 100%, the kth measurement is k*100/n% of the total time. Acceleration was determined for the percentage time points of the exercise from 0.5 every 5 to 100 using the interpolation method."

Instead of drawing yellow line it's better to highlight the contact in another alternative way rather than drawing line as it disturbs the average line graph. 

The recommended changes have been made

Kindly choose only one either table or figure as the information is almost similar. The important p values you can mention in the results section. In my opinion the rest of the results can be provided as supplementary document. 

The recommended changes have been made and the charts have been included in the supplementary document. This is due to the fact that this very article very often refers to a large number of data from tables in its descriptive part.

  Kindly provide limitation of the study and possible future studies. 

Changes have been made to the lines 510-517

For a full analysis of the accelerations acting on the head during the contact of the buttocks with the ground, the component of acceleration in the frontal plane should be taken into account, which may constitute the subject of the next publication. However, preliminary analyses show that there are no differences between the studied groups in this phase of the fall. It would be interesting to analyze the acceleration in the frontal plane to estimate its value at the moment of stopping the board on which the participant is standing, just before the fall, as in the first phase. It would be interesting in terms of biomechanical interpretation.

Changes have been made to the lines 571-578

The results presented in this article diagnose the susceptibility to head, neck and pelvic injuries during a fall. It would be beneficial to increase the number of sensors attached not only to the head, especially when determining the susceptibility to pelvic damage. Attempts were made in this regard, but none of the results obtained were as accurate as those obtained from the sensor on the head, due to the problem of attaching the sensors to other body parts. The sensor on the head was pressed against the forehead with a band that adequately limited the possibility of its movement in the course of the tests.

The items 28--33 have been added to the literature

28 -  Court-Brown C, Heckman J, McQueen M, Ricci W, McKee M. Rockwood and Green’s Fractures in Adults. 8th Edition. Lippincott; 2015.

  1. Tile M. Acute Pelvic Fractures: I. Causation and Classification. J Am Acad Orthop Surg. 1996;(4):143-151. doi:10.5435/00124635-199605000-00004
  2. Bailey R, Abernathy B, Lisa K, et al. Low-Energy Pelvic Ring Fractures: A Care Conundrum. Geriatric Orthopaedic Surgery & Rehabilitatio. 2021;(12):1-7. doi:10.1177/2151459320985406
  3. Fuchs T, Rottbeck U, Hofbauer V, Raschke M, Stange R. Pelvic ring fractures in the elderly. Underestimated osteoporotic fracture. Unfallchirurg. 2011;(114):663-670. doi:10.1007/s00113-011-2020-z
  4. Leslie M, Baumgaertner MR. Osteroporotic pelvic ring injuries. Orthop Clin North Am. 2013;(44):217-224.
  5. Oberkircher L, Ruchholtz S, Rommens PM, Hofmann A, Bücking B, Krüger A. Osteoporotic pelvic fractures. Dtsch Arztebl Int. 2018;(115):70-80. doi:10.3238/arztebl.2018.0070

Removed :

Cynarski W.J., Sieber L., Szajna G. Martial arts in physical culture. Ido Movement for Culture Journal of Martial Arts Anthropology. 2014;14(4):39-45. doi:10.14589/ido.14.4.5

Reviewer 2 Report

The paper presents the analysis of transverse acceleration measurements from two groups of sportsmen at two levels of backwrds fall, IFT and FFT. 

Since accelerations in Newtonian mechanics are related to forces, these measurements directly indicate the level of possible head injury.

The two groups of sportsmen are from two different disciplines where falls are frequent. The persons in the two groups on average are relatively similar in body size, thus the measurements should display similar effects. Thus, selection of sample people are okay for the study.

The authors computed average accelerations at various stages of a fall. The statistics of the results are given in tables.

It is found that sportsmen trained in martial arts consistently fall better than handball players in IFT and FFT, which is indicated by lower standard deviations.  Hence, a proper training for falling techniques can reduce head or spine inuries.  

The methods are sicientifically rigorous and the results are meaningful with useful indications for the readers. 

The text has several broken sentences, which are marked in the attached file. Otherwise the use of English is correct.

The authors collected acceleration data from persons forced to fall backwards on a turn table. The subjects (persons) pertain to two groups of professional sportsmen with different motor reactions to falling. The paper tries to compare the efficiency of their falling skills acquired through training in two different performance in sports, handball and martial arts. It is observed that the authors made special efforts to make measurements as unbiased as possible; eg., they choose sportsmen with similar body sizes (mass and length) and they eliminated those who support the body by the hand before impact with the ground, the "hand error" as they call it. The data comprises one point 3 component linear accelerations of the head. The authors think that the injury should be proportional to the level of accelerations, which is plausible, but not fully correct especially if only one component of acceleration is considered. The authors are not engineers, thus within the limits of their measurement capabilities as well as knowledge of mechanics the authors tried to make the data useful by some statistical analyses and based their discussions on them. (1) How do you interpret the high (about 2g) initial accelerations in group A ? The average accelerations, in group B, however, are below 1g at the begining of a fall. Why group A's reaction to fallin gives higher initial accelerations ? (2) Please give two examples of full acceleration time histories for each of groups A and B. And, discuss the various stages of a fall (like, buttocks hitting ground etc) on these graphs. (3) I think a velocity time history would also be useful in interpreting impact and duration of impact of buttocks on the ground. (4) Please discuss the significance of component of acceleration to head injury, referring your previous article using sagittal plane accelerations. Which component of acceleration relates more to the injury ? (5) Do you have suggestions as to the technique of falling beyond just backward roll or side aligning of the body to be acquired by sportsmen to reduce injuries ? (6) The accelerometer has 3 components, so what about the frontal plane accelerations ? How would they ralte to the results ? (7) Single point of measurement of accelerations does not give information about the rotations of the body. For a better study of falling body rotations must also be measured. What do you suggest to compute or measure rotations of the torso as a future work ? (8) The study comprises only accelerations. No force measurements were made, nor computed. Knowing the head accelerations and making simplifying assumptions, like torso comes to full stop upon buttock hit the ground, the normal and shear forces and couples that developed in the neck can be computed. These forces would be a better indication of impending neck injury and thus can be used in discussions. (9) Injuries can also be inflicted on the pelvis when buttocks hit the ground. Can you discuss how this type of injury be estimated ?

Author Response

Response to Reviewer 2  Comments

All changes made to the text (manuskrypt) recommended by the reviewers are highlighted in yellow. In the article all changed sentences are marked in yellow color.

. (1) How do you interpret the high (about 2g) initial accelerations in group A ?

The average accelerations, in group B, however, are below 1g at the begining of a fall. Why group A's reaction to fallin gives higher initial accelerations ?

We do not interpret the 2g value as particularly high as the beginning of the measurement of linear acceleration in the transverse plane given by the sensor it started at the value of 1g. This information is included in the line (285 - 289.

‘In the statistical calculations, the acceleration value in m/s2 from the sensor was considered, taking into account the gravitational acceleration g. This means, for example, that the given value of 0.1 acceleration means 0.1 g (0.1 x 9.81 m/s2 = 0.981 m/s2). The beginning of the measurement of linear acceleration in the transverse plane given by the sensor started from the value of 1g.’

This does not affect the significance of statistical studies. In fact, the average acceleration in group A is 1g. As further justification, we will state that we have checked that in a freely falling trampoline jumper, before landing on the net with his feet, the sensor mounted on his forehead indicates 0 g. This may additionally explain the achievement of values less than 1g in group B.

 Extra additions in lines 491 - 509

It should be noted that for the IFT at time points from 0 to 20, the standard deviations in group A are larger than in group B (Table 7), and for the FFT at time points from 0 to 15 (Table 8). At the same time, the positive and negative values of the average acceleration of the group (Tables 1 and 2) and the selected student (Fig. 3 and. 4) at the beginning of the 1st stage of the fall are higher in group A compared to group B.

Inverse relationships between the groups are observed in stage 2 of the fall. The explanation of these observations may be offered by the way of performing falls in the groups (Fig 2a and 2b). Obtaining greater accelerations in group A in the first stage of the fall may lead to transferring ground reactions to the head through the trunk and lower limb as a result of rolling on it. This reaction is especially visible when the lower limb is withdrawn and the entire load is transferred through the foot at the beginning of this stage [21]. Performing these motor activities requires the acquisition of certain motor habits. The analysis of this phase of the fall would require extending the research for the fall technique used in group A. In group B, the fall technique in this phase resembles the behaviour of a vertical pole once it has been fallen. One can imagine there is a head at the top of the pole. It is obvious from the mechanical perspective that the acceleration of the head in the transverse plane indicated by the sensor will be less than 1g. This may account for the assumption of lower head acceleration values  in stage 1 of the fall in group B.

 (2) Please give two examples of full acceleration time histories for each of groups A and B. And, discuss the various stages of a fall (like, buttocks hitting ground etc) on these graphs.

They were taken in Fig 3 and 4

An overview is included in lines 243 - 257.

The entire fall sequence in both groups was divided into three time stages: 0% ≤ stage 1 < 40%, 40% ≤ stage 2 ≤ 60%, 60% < stage 3 ≤ 100%. In this article the stages of the fall sequence are marked in the graphs and tables provided in the results section. In stage 1 there are the greatest differences in the performance of the fall between the groups. In group A, the trainee withdraws one leg transferring loads to it through the foot in order to set the body to the lateral position, then rolls through the lower leg, thigh to the buttocks (Fig 2a) [10,11,21], while in group B (Fig. 2b), after losing balance the transition is made to the contact with the buttocks. During this time, the trainee should change their body segment alignment, leading to the appropriate adjustment of the angles β, γ and ε (Fig. 1) that shall take place during the collision with the ground [14,25]. In stage 2 in both groups there is contact of the buttocks with the ground. Stage 3 of falling backward is the most common contact in both groups of the trainees: i.e. of the back with the ground. The analysis of the acceleration readings was halted when there was no danger of the head contact with the ground. This was usually the case when the head was parallel to the ground (Figure 2c).

(3) I think a velocity time history would also be useful in interpreting impact and duration of impact of buttocks on the ground.

A very good idea, but rather for the following article. This was not done for the sensor indications for acceleration in the sagittal plane, and there would be no comparisons of the results to the transverse plane. 

(4) Please discuss the significance of component of acceleration to head injury, referring your previous article using sagittal plane accelerations. Which component of acceleration relates more to the injury ?

We are unable to accurately answer this question as to value accelerationes  . It depends on the angles (ß, γ, ε) shown in Fig 1. This answer can also be obtained from analysing lines135 -145,

The above analysis suggests that vector X' will be closest to the direction of linear acceleration in the sagittal plane aS, and vector N for the transverse plane aT. This analysis shows that the indications of aT acceleration will be the most informative in terms of the risk of the head, neck, spine and then pelvic injury as a result of the connection with the N force. On the other hand, the as indications about the risk of hitting the head during a fall due to the connection with the X' force. Simultaneously, it should be noted that the discussed accelerations indicated by the sensor may be affected by the participation of the head in the curvilinear movement of gravitational fields. It is difficult to find scientific studies on the subject of falling backwards. Moreover, the accelerations achieved may be influenced by the muscle force generated by the players and acting on the head during the fall.

 (5) Do you have suggestions as to the technique of falling beyond just backward roll or side aligning of the body to be acquired by sportsmen to reduce injuries ?

The answer is in lines 535-543

Falling backwards using a technique similar to the gymnastic backflip will be justified when the resultant force causing the fall depends mainly on the vertical component. This occurs mainly in players practicing disciplines where jumps in the vertical direction are frequent, i.e. volleyball, basketball, handball [25]. After completing education and training, a large part of the populationno longer practices such sports. A frequent cause of a fall is a slip, which is dominated by the horizontal component of the force inducing the fall [25,40]. It is a frequent occurrence in everyday physical activity, especially in the elderly. During everyday movement, practicing vertical jumps is rare, especially in the elderly.  

And line 550 - 554

It follows from the above considerations that since horizontal force is a frequent cause of falls, teaching the technique of falling backwards with a lateral body position should be included in the school curriculum [11]. At the same time, in order for the movement habits to consolidate, it is recommended to practice this fall technique throughout one’s life as part of one’s own physical safety.

(6) The accelerometer has 3 components, so what about the frontal plane accelerations ? How would they ralte to the results ?

The answer is in lines 510-514

For a full analysis of the accelerations acting on the head during the contact of the buttocks with the ground, the component of acceleration in the frontal plane should be taken into account, which may constitute the subject of the next publication. However, preliminary analyses show that there are no differences between the studied groups in this phase of the fall.

 (7) Single point of measurement of accelerations does not give information about the rotations of the body. For a better study of falling body rotations must also be measured. What do you suggest to compute or measure rotations of the torso as a future work ?

The answer is in lines 514 -517

It would be interesting to analyze the acceleration in the frontal plane to estimate its value at the moment of stopping the board on which the participant is standing, just before the fall, as in the first phase. It would be interesting in terms of biomechanical interpretation.

(8) The study comprises only accelerations. No force measurements were made, nor computed. Knowing the head accelerations and making simplifying assumptions, like torso comes to full stop upon buttock hit the ground, the normal and shear forces and couples that developed in the neck can be computed. These forces would be a better indication of impending neck injury and thus can be used in discussions.

In order to answer this question, the "Introduction" has been extensively expanded - the explanation in lines.  The room for future biomechanical publications is extensive here.  One of the authors of this article is a physicist.  However, we try to present the mechanics in the most accessible way. At the same time, there are few publications on the biomechanical analysis of a backward fall. It is true that no force measurements have been made - this can be done in the future using additional, properly mounted sensors. At the same time, it will be necessary to consider the weight of the respective body segments.  At the same time, the problem is that in these tests there is a movement of the subject in the 2nd and 3rd stage of the fall, and there is no complete halt. Still in group B the movement is sometimes restricted.  The issue of measuring forces during a fall is interesting but requires further theoretical refinement and expansion of the equipment base. The measurement of forces is difficult as the acceleration itself depends on other factors as indicated in the lines.

Extensive changes have been made here.  The editing was accopanied by addition to the background and the introduction of an extended biomechanical description with an explanation of new Fig 1.

Explanations in lines 108 - 144

In order to understand it better, it is necessary to extend the considerations of Mroczkowski, who presented the biomechanical analysis of the moment of force acting on the head during a backward fall [14]. Based on his assumptions, this article presents an analysis of the relationship between the indications of the sensor mounted on the head of the falling person (Fig 1b), in terms of the direction and sense of the linear acceleration vectors in the transverse and sagittal planes, with these features of the force vectors acting on the head. Using the F = ma relation resulting from Newton's second law of motion, it is obvious that the direction and sense of the force and acceleration vectors are the same with the values being directly proportional. At the same time, the sensor, due to its stable attachment and negligible mass, can be considered as one solid body with the head.

Mroczkowski concluded that the torso and head may be considered as two links of the kinematic chain, recognizing the connection between them as articulated (Fig. 1), which in the event of a moment of force causes the head to move [14]. For this reason, it is assumed that there is a vector P corresponding to the weight of the head. If all the bodies were treated as solid during the collision, the considerations would assume that the W and R vectors should differ only in their directions. At the same time, by projecting the vector W onto the line connecting the designated point of contact of the buttocks with the ground and the centre of gravity of the head, the N vector (Fig 1b) is obtained (marked with a dashed line, as an exception). The N vector should also be equal to the resultant vector that would be created from the addition of vectors P and Y'. The N vector should only differ from the Y vector in sense. The same relationships will occur if the sensor indications were analysed in the other settings during the fall (Fig 1a,c,d). Some discrepancies from the relationships described above may result from the fact that the value of the P vector was adopted without the precise determination of the head mass. Its length may therefore be inconsistent with reality. In reality, the change of the other vectors marked in the centre of gravity of the head may be affected by the fact that not all the energy during the impact with the ground will be transferred to the head [14].The above analysis suggests that vector X' will be closest to the direction of linear acceleration in the sagittal plane aS, and vector N for the transverse plane aT. This analysis shows that the indications of aT acceleration will be the most informative in terms of the risk of the head, neck, spine and then pelvic injury as a result of the connection with the N force. On the other hand, the as indications about the risk of hitting the head during a fall due to the connection with the X' force. Simultaneously, it should be noted that the discussed accelerations indicated by the sensor may be affected by the participation of the head in the curvilinear movement of gravitational fields. It is difficult to find scientific studies on the subject of falling backwards. Moreover, the accelerations achieved may be influenced by the muscle force generated by the players and acting on the head during the fall.

(9) Injuries can also be inflicted on the pelvis when buttocks hit the ground. Can you discuss how this type of injury be estimated ?

The answer is in lines 145-160

The biomechanical analysis presented in Figure 1 shows that when the buttocks hit the ground, the main component of the force acting on the pelvis during the injury is pointed upwards (Y), with the smaller component (X) directed forwards or backwards. In the Young-Burgess Classification of pelvic fractures [28,29] it is the VS (Vertical Shear) type. With this type of fracture, the pubic and ischial bones limiting the obturator foramen may be fractured, which carries the risk of damage to the vascular and nervous structures passing through this hole and damage to the lesser pelvic organs.

The age group particularly at risk of fractures, including pelvic fractures, are the elderly, who suffer from a decrease in bone mineral density and weakening of the mechanical strength of the bones associated with osteoporosis. Pelvic fractures due to a backward fall can be classified as a low-energy fractures, particularly characteristic of the elderly population. According to scientific reports, regardless of the classification into fall types, falls among people over 65 years of age are the most common cause of such fractures [30,31]. There is an increase in the incidence of low-energy pelvic fractures in people with osteoporosis [30,32]. Osteoporotic pelvic fractures in the elderly result in an annual mortality rate of 9.5% to 27% among the elderly[33].

Reviewer 3 Report

This manuscript investigated the changes in linear acceleration of the head in the transverse plane during a backward fall forced by IMU sensors. Some modifications need to be done before it can be published.

1. Fig. 1-4 have been used in the Ref 11 by the authors.

2. The paper is lengthy, some data Tables can be put as supporting materials.

Author Response

Response to Reviewer 3  Comments

Review Report Form

Open Review

Quality of English Language

( ) English very difficult to understand/incomprehensible
( ) Extensive editing of English language and style required
( ) Moderate English changes required
(x) English language and style are fine/minor spell check required
( ) I am not qualified to assess the quality of English in this paper

Yes

Can be improved

Must be improved

Not applicable

Does the introduction provide sufficient background and include all relevant references?

(x)

( )

( )

( )

Are all the cited references relevant to the research?

(x)

( )

( )

( )

Is the research design appropriate?

(x)

( )

( )

( )

Are the methods adequately described?

(x)

( )

( )

( )

Are the results clearly presented?

(x)

( )

( )

( )

Are the conclusions supported by the results?

(x)

( )

( )

( )

Comments and Suggestions for Authors

This manuscript investigated the changes in linear acceleration of the head in the transverse plane during a backward fall forced by IMU sensors. Some modifications need to be done before it can be published.

  1. 1-4 have been used in the Ref 11 by the authors.

Response : thank you very much for your remark. Change have been made by putting some figures in Appendix S1.

  1. The paper is lengthy, some data Tables can be put as supporting materials.

Response : thank you for your remark. We agree with this comment and we decreased the length of the paper by using Appendix S1

Submission Date

18 January 2023

Date of this review

19 Feb 2023 04:46:05

Round 2

Reviewer 1 Report

Thank you for addressing all the questions and providing answers. Could you kindly review the manuscript and ensure consistency throughout, such as using "Figure" or "Fig" according to the author's guidelines before submitting the final version?

Author Response

Response to Reviewer 1  Comments

REVIEWER 1

Author's Reply to the Review Report (Reviewer 1)

Please provide a point-by-point response to the reviewer’s comments and either enter it in the box below or upload it as a Word/PDF file. Please write down "Please see the attachment." in the box if you only upload an attachment. An example can be found here.

* Author's Notes to Reviewer

p

Word / PDF

or

Haut du formulaire

Review Report Form

Open Review

Quality of English Language

( ) English very difficult to understand/incomprehensible
( ) Extensive editing of English language and style required
(x) Moderate English changes required
( ) English language and style are fine/minor spell check required
( ) I am not qualified to assess the quality of English in this paper

Yes

Can be improved

Must be improved

Not applicable

Does the introduction provide sufficient background and include all relevant references?

(x)

( )

( )

( )

Are all the cited references relevant to the research?

(x)

( )

( )

( )

Is the research design appropriate?

( )

(x)

( )

( )

Are the methods adequately described?

( )

(x)

( )

( )

Are the results clearly presented?

( )

(x)

( )

( )

Are the conclusions supported by the results?

( )

(x)

( )

( )

Comments and Suggestions for Authors

Thank you for addressing all the questions and providing answers. Could you kindly review the manuscript and ensure consistency throughout, such as using "Figure" or "Fig" according to the author's guidelines before submitting the final version?

Response : thank you very much for your remark. Recommendations have been implemented.

All comments for corrections are marked in the manuscript.

Submission Date

18 January 2023

Date of this review

10 Mar 2023 10:00:27

Bas du formulaire
